# Preparation and Properties of Fractionated Soybean Protein Isolate Films

**DOI:** 10.3390/ma14185436

**Published:** 2021-09-20

**Authors:** Yunxiao Wei, Ze’en Huang, Zuolong Yu, Chao Han, Cairong Yang

**Affiliations:** 1Biology and Environment Engineering College, Zhejiang Shuren University, Hangzhou 310015, China; lvwyx@163.com (Y.W.); hanchao96@163.com (C.H.); a15957547059@163.com (C.Y.); 2Jiangsu Province Key Laboratory of Fine Petrochemical Engineering, Changzhou University, Changzhou 213164, China; hze@cczu.edu.cn

**Keywords:** soybean protein isolate, packaging film, fractionated separation, mechanical property, permeability

## Abstract

Soybean protein isolate (SPI) and its four fractionated products (7S globulin, 11S globulin, upper soybean residue, and lower soybean residue) were compared by fabricating films and film liquids. The separation and grading effects, rheological properties of the film liquids, and difficulty in uncovering the films, in addition to the mechanical properties, water vapor permeability, oil permeability, and surface morphology of the films, were investigated. Results showed that the centrifugal precipitation method could be used to produce fractionated products. The 7S and 11S globulin films exhibited better hydrogels at lower shear rates than the other SPIs; however, they were more difficult to uncover. The tensile strength of the graded films decreased by varying degrees. However, the elongation at the break of the upper soybean residue film considerably increased, reaching 70.47%. Moreover, the permeability and surface morphology of the film were enhanced or weakened. The fractionated products, 7S and 11S globulin films, exhibited better performance. Overall, this study provides a basis for the improved development and use of fractioned SPI products.

## 1. Introduction

Edible packaging films have recently become a research hotspot because they are environmentally friendly and can be used in food packaging. Currently, the raw materials for preparing edible packaging films mainly include polysaccharides, proteins, and lipids [1,2,3,4]. Owing to the different raw materials used in packaging, the performance of different packaging films differs. For example, packaging films based on cellulose, starch, and other polysaccharides show the characteristics of oil resistance, good transparency, and poor mechanical properties. However, packaging films based on proteins exhibit good strength and elasticity, as well as antibacterial properties [5,6]. In addition, protein composite films synthesized using different raw materials show improved film performance and enhanced applications [7,8].

The interaction of intramolecular and intermolecular hydrophobic bonds and disulfide bonds form the network structure of soybean protein isolate (SPI) films, which exhibit low gas permeability, excellent mechanical properties, and poor barrier performance [9,10,11]. The protein content in SPI exceeds 90% and mainly constitutes 2S, 7S, 11S, and 15S globulin. Different proteins show different emulsifying abilities and gel strengths [12,13]. Thus, the purification and yield of obtained products vary considerably based on different separation and grading processes of soybeans [14,15,16,17].

Fractioned SPI products are mostly used in the food industry. Chiba tofu can be prepared with different degrees of hardness and elasticity by adjusting the ratio of 7S to 11S globulin [18,19]. The foaming properties of SPI are improved using the combined preheat treatment and controlled enzymatic hydrolysis to change the 7S and 11S globulin ratio [20]. Based on the comparison of the hepatoprotective effect of SPI, 7S globulin, and 11S globulin, 7S shows optimal performance in food processing [21]. Proteins are also directly used in food packaging and coating technology owing to their biodegradability, processability, combination positions, and nontoxicity to food [22,23]. In this study, the separation method of graded SPI, which was optimized by Chong [24], was used to fabricate four products, and the properties of their edible packaging films were compared for precise applications.

## 2. Materials and Methods

### 2.1. Preparation of Fractionated SPI Products

The specific steps for preparing fractionated SPI products based on the literature [24] are as follows: 24 g SPI (SOYGLOW, Pingdingshan, China) was mixed with 360 mL water. The pH was adjusted to 8 using 2 mol/L sodium hydroxide solution, and the mixture was mechanically stirred for 1 h. The obtained solution was centrifuged (TDL-5-A, Shanghai Anting Scientific Instrument Factory, Shanghai, China) at 200× *g* for 10 min to obtain a precipitate, i.e., the lower SPI residue, and the filtered cake was further filtered using a 200 mesh sieve to obtain the upper SPI residue. To prepare the 11S globulin extract, 3 mM sodium sulfite and 5 mM anhydrous magnesium chloride were added to the 11S globulin extract. Further, the pH was adjusted to 5.5 using 2 mol/L HCl solution; then, the mixture was stirred at room temperature for 10 min and centrifuged at 200× *g* for 10 min. After centrifugation, the pH of the protein extract was adjusted to 4 using a 2 mol/L HCl solution while stirring for 10 min and centrifuging at 200× *g* for 10 min. The obtained precipitated protein was 7S globulin. The two globulin products were then redispersed using deionized water, and the pH of both solutions was adjusted to 7.5 using a 2 mol/L HCl solution. The final product was obtained by freeze-drying, and four fractionated SPI products were finally obtained.

### 2.2. Characterization of Fractionated SPI Products

Based on the literature [25] and a slight modification, 4 mg samples (SPI, 11S, 7S, lower protein, and upper protein) were weighed and dissolved in a 1 mL phosphate-buffered solution. After full vortex mixing at 2000× *g*, 20 μL of the mixed solution was combined with 5 μL of loading buffer 5X and heated at 100 °C for 5 min. The samples were prepared and subjected to sodium dodecyl sulfate–polyacrylamide gel electrophoresis (SDS-PAGE) (JY-ECP3000, Beijing JUNYI Electrophoresis Co., Ltd., Beijing, China). SDS-PAGE was performed at a constant voltage of 150 V using 4–15% of a denaturing electrophoresis precast glue and stained with Coomassie brilliant blue, decolorized, and scanned.

### 2.3. Preparation of SPI Films

Five films were prepared using 10 g of SPI or fractioned SPI products as the matrix. As additives, 0.2 g sodium carboxymethyl cellulose (Shanghai Aladdin Biochemical Technology Co., Ltd., Shanghai, China), 0.12 g carrageenan (Tengzhou Tongda Algae Engineering Technology Co., Ltd., Tengzhou, China), 0.1 g glycerol monodistearate (Shanghai Aladdin Biochemical Technology Co., Ltd., Shanghai, China), and 1.5 g glycerol (Shanghai Lingfeng Chemical, Ltd., Shanghai, China) were added to 100 mL deionized water. The pH was adjusted to 8 using 0.5 mol/L sodium hydroxide, and the mixture was stirred at 300 rpm for 30 min at 80 °C. After vacuum degassing, the film liquids were rolled onto a steel plate, dried in an oven at 60 °C, uncovered, and stored for further use.

### 2.4. Characterization of SPI Films

#### 2.4.1. Rheology

The apparent viscosities and shear rate curves of the film liquids were determined using a rheometer (MCR 102, Anton-Paar, Shanghai, China). Measurements were performed at a constant temperature of 80 °C, using a conical plate with a diameter of 50 mm and an angle of 1° and the flow curve measurement mode. The shear rate was considered as a variable with a range of 0.01–100 s^−1^, the mode was linear scan, and 50 variable points were selected.

#### 2.4.2. Contact Angle

The static contact angle of the fractionated film solutions was measured at 80 °C to characterize their affinity with the coated sheet. The contact angle between the film solution and the coating plate was measured using a camera device (JC2000D3, Shanghai Zhongchen Digital Technic Apparatus Co., Ltd., Shanghai, China).

#### 2.4.3. Mechanical Properties

The sample was cut into strips with a length of 10 cm and a width of 0.5 cm. The tensile strength (*T_S_*) and elongation at the break (*E*) of the films were measured (TA.XT Plus, Stable Micro System, London, Britain) along the vertical and horizontal directions using three strips. In total, six parallel samples were measured. The standard distance was 50 mm, and the sampling speed was 100 mm/min.
(1)$$T_{S}=F/S,$$
where *T_S_* is the tensile strength (MPa), *F* is the tensile force of the film (N), and *S* is the cross-sectional area of the film (m^2^).
(2)$$E=(L-L_{0})/L_{0}\times 100\%,$$
where *E* is the fracture elongation (%), *L*_0_ is the standard distance (50 mm) of the sample (mm), and *L* is the elongated distance of the specimen fracture (mm).

#### 2.4.4. Fourier Transform Infrared (FTIR) Spectroscopy

A small piece of the fractionated SPI film samples was cut and dried. Then, the samples were mixed with KBr to be pressed into tablets. Fourier transform-infrared (FTIR) spectroscopy (FTIR-650, Bruker, Karlsruhe, Germany) was performed to investigate the influence of the film composition on the film structure. The FTIR spectra were obtained by scanning 64 times at room temperature at a resolution of 4 cm^−1^ and the wavenumber range of 500–4000 cm^−1^.

#### 2.4.5. Permeability

In a 100 mL beaker, 50 g anhydrous calcium chloride (particle size of 2 mm) was added. Uniform smooth films without holes or wrinkles were selected. Subsequently, they were measured to determine their thicknesses, sealed at the mouth with molten paraffin, placed in a dryer with 100% relative humidity, measured at 25 °C, and removed from the dryer, and weighed every 24 h. Three parallel experiments were continuously performed for one week, and the results were presented as the arithmetic mean of each group.
(3)WVP=Δm×d/(A×t×ΔP),
where *WVP* is the water permeability coefficient (g·mm/m^2^·d·KPa), Δ*m* is the steady mass increment (g), *d* is the film thickness (mm), *A* is the effective measured area (m^2^), *T* is the time interval (days) of the measurement, and Δ*P* is the vapor pressure difference (KPa) on both sides of the sample.

The film sealing was measured by placing approximately 5 mL salad oil in a test tube and then sealing it with the film. The oil was further placed upside down on a filter paper for a week to calculate the average oil permeability coefficients of the three samples.
(4)Poil=Δm×d/A×T,
where *P_oil_* is the oil permeability coefficient (g·m/m^2^·d), Δm is the change in the filter paper quality (g), *d* is the film thickness (mm), *A* is the film area (m^2^), and *T* is the placement time (days).

#### 2.4.6. Scanning Electron Microscopy (SEM)

The surface morphologies of the PSI films were sprayed with gold and observed using scanning electron microscopy (SEM; S-570, Hitachi, Ltd., Tokyo, Japan) at a working voltage of 10 kV. Images of the samples were captured after spraying them with gold.

## 3. Results and Discussion

### 3.1. Analysis of the Grading Effect

Fractioned SPI products were obtained using various purity and yield values (Figure 1 and Figure 2 and Table 1) [26]. The lower protein was not dissolved under this electrophoresis condition; however, similar bands were observed for the other samples. This was primarily because different proteins contained different subunits and exhibited different physicochemical properties, while the α′, α, and β subunits of 7S and the acidic and basic subunits of 11S globulin were observed [27,28]. The SPI showed strong 7S and 11S globulin compositions. The upper protein yield was different owing to variations in the separation experiments, and there were no products even after modifying the experimental conditions. As more than 90% of SPI is proteins whose spatial structures and amino acid compositions directly affect the physicochemical properties of protein isolates, the selection of separation methods is crucial [29].

### 3.2. Rheology

Figure 3 shows the rheological properties of each film solution. The viscosity gradually decreased with the increasing shear rate. However, the 11S globulin film solution exhibited a large mutation, and an inflection point was observed when the shear rate was 0.241 s^−1^. Furthermore, the viscosity of this film solution was lower than that of the SPI film solution. This indicated that the 11S globulin film solution shows a shear-thinning phenomenon and is a non-Newtonian fluid. The gel and emulsification properties of the 7S and 11S globulin film solutions were better than those of the SPI film solution; thus, they achieved high viscosity values at low shear rates. However, the shear resistance of the globular protein decreased at high shear values [30].

### 3.3. Contact Angle

The contact angle between the film solution and the coating plate indicates the difficulty in uncovering the film after its formation [31]. Based on the analysis of the contact angle of the film solutions of SPI and fractioned products (Table 1 and Figure 4), a larger contact angle indicated that the film was easier to uncover. The main reason for differences in the contact angles of the samples was that the hydrophilicity values of different graded proteins were different after gelation and the extent of adhesion to the steel plate was different. For example, 7S globulin had more polar amino acids and showed a smaller molecular gap than others after mixing with additives and close contact with the steel plate [32].

### 3.4. Mechanical Properties

Figure 5 depicts certain differences in the mechanical properties of the films of SPI and fractioned products. For example, the *T_S_* value decreased from 12.94 MPa (SPI film) to 1.22 MPa (upper protein film), and the *E* value increased from 2.91% (SPI film) to 70.47% (upper protein film). The upper protein film contained more small components than SPI, which promotes plasticization once the film is prepared, allowing molecules to slide more easily and enhancing fracture elongation. However, the mechanical properties of the 7S and 11S globulin films were slightly different. This is primarily because the polar amino acids in proteins promote the plasticizing effect of hydrophilic additives; however, proteins with small molecular weights cannot form complex spatial winding networks, resulting in lower *T_S_* than SPI films.

### 3.5. FTIR

Figure 6 shows the FTIR spectra of the five sample films. The experimental results showed similar characteristic absorption peaks of the sample membranes, such as amide I (C=O stretching vibration), amide II (N–H bending vibration), and amide III (C–N and N-H stretching vibrations) of the corresponding proteins at 1666, 1538, and 1233 cm^−1^, respectively [33]. However, the characteristic absorption peaks at 2919 cm^−1^ (=C-H and -NH^3+^ asymmetric stretching vibrations) and 1045 cm^−1^ (C-H and C-O-H stretching vibrations) were considerably different in terms of the intensity, indicating that the 7S and 11S globulin films had more exposed polar bonds to produce hydrogen bonds. FTIR spectrum analysis further revealed that the precipitation separation approach distinguished the molecular weight of proteins and altered the conformation of the α-helical and β-folding proteins, resulting in unique characteristics of the products.

### 3.6. Gas Permeability

The permeability of the SPI film was considerably affected after grading (Figure 7). The water vapor permeability of the 11S globulin film was lower than that of the SPI film, whereas those of the other graded protein films were higher than that of the SPI film. However, the oil permeability of the graded protein films decreased, compared with the SPI film, with the 11S globulin film showing a decrease from 78.76 to 5.82 g·m/(m^2^·d). As permeability was determined based on the molecular gap and amphiphilicity of membrane components, the permeability of the gas molecules was determined based on the permeation speed [34]. The SPI films with different molecular weights formed a dense, crosslinked structure, which could effectively prevent the passage of water vapor, although their lipophilicity tendency was stronger than those of the other films.

### 3.7. Surface Morphology

Table 1 and Figure 8 demonstrate that the surface morphologies of the various protein films varied considerably. The upper and lower protein films were coarser than the SPI films, with obvious folds and differently sized bulges. The surface morphologies of the 7S and 11S globulin films were smooth and without banded bulges. The poor solubility of the upper and lower proteins mainly resulted in uneven surface morphologies. However, the 7S and 11S globulins with good homogeneity showed excellent emulsification and gelling properties after gelation, which is beneficial for the film extension.

## 4. Conclusions

In this study, four fractioned SPIs were synthesized using the optimized centrifugation method. The products exhibited different characteristics from the raw materials. Comparing the film properties of the SPIs and fractioned products, the viscosities of the 7S and 11S globulin film solutions were higher at low shear rates than those of the SPI film solution; however, they rapidly decreased at high shear rates, indicating a non-Newtonian fluid type. The film components with single molecular weights and high hydrophilicity were more difficult to uncover from the steel plate. The 11S globulin film showed reduced water vapor and oil permeabilities, from 245.16 and 78.76 to 144.45 and 5.82 g·m/(m^2^·d), respectively. This study indicates that SPI grading can produce food packages or functions for precise development and use.

## Figures and Tables

**Figure 1 materials-14-05436-f001:**
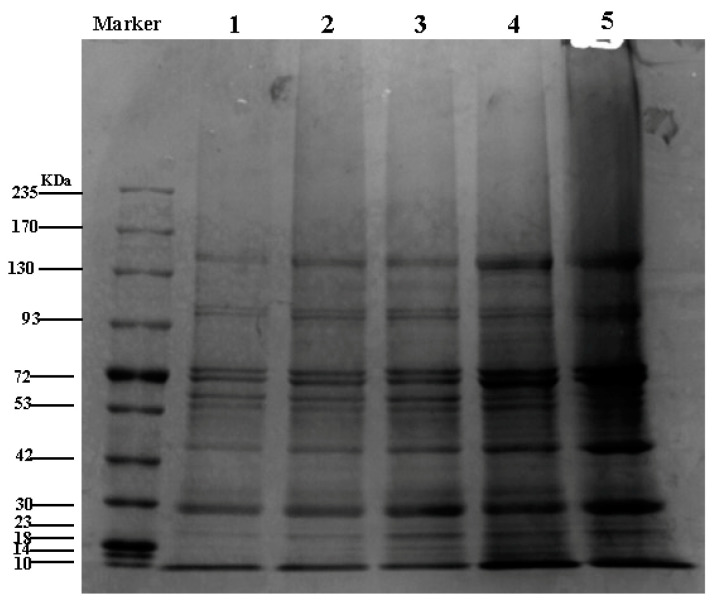
SDS–PAGE of SPI and fractioned products. Marker: the different molecular numbers of proteins: (1) lower protein, (2) upper protein, (3) 11S globulin, (4) 7S globulin, and (5) SPI.

**Figure 2 materials-14-05436-f002:**
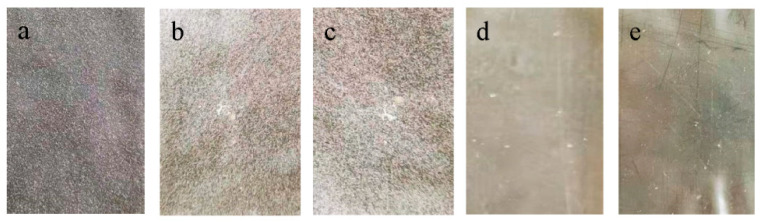
Films of SPI and fractioned products: (**a**) SPI, (**b**) lower protein, (**c**) upper protein, (**d**) 11S globulin, and (**e**) 7S globulin.

**Figure 3 materials-14-05436-f003:**
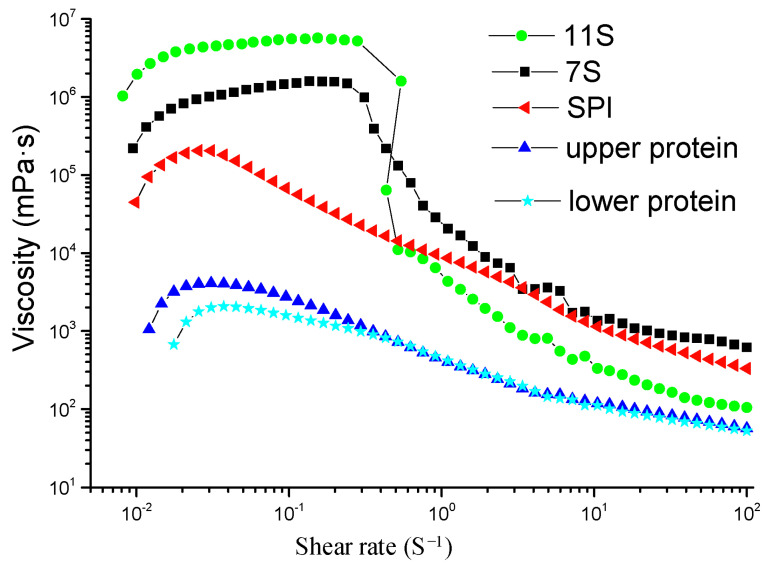
Rheological properties of the film solution of SPI and fractioned products.

**Figure 4 materials-14-05436-f004:**
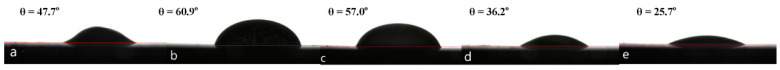
Contact angle of the film solutions of SPI and fractioned products: (**a**) SPI, (**b**) upper protein, (**c**) lower protein, (**d**) 11S globulin, and (**e**) 7S globulin.

**Figure 5 materials-14-05436-f005:**
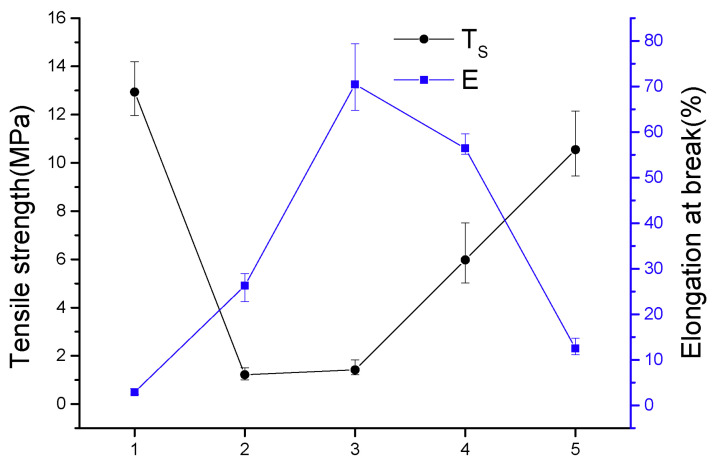
Mechanical properties of the film solutions of SPI and fractioned products: (1) SPI, (2) lower protein, (3) upper protein, (4) 11S globulin, and (5) 7S globulin.

**Figure 6 materials-14-05436-f006:**
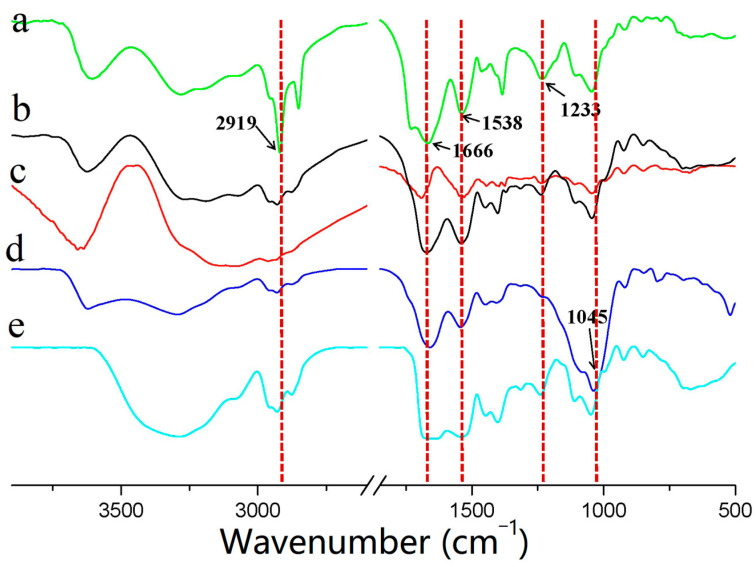
FTIR spectra of the film solutions of SPI and fractioned products: (**a**) 7S globulin, (**b**) SPI, (**c**) upper protein, (**d**) lower protein, and (**e**) 11S globulin.

**Figure 7 materials-14-05436-f007:**
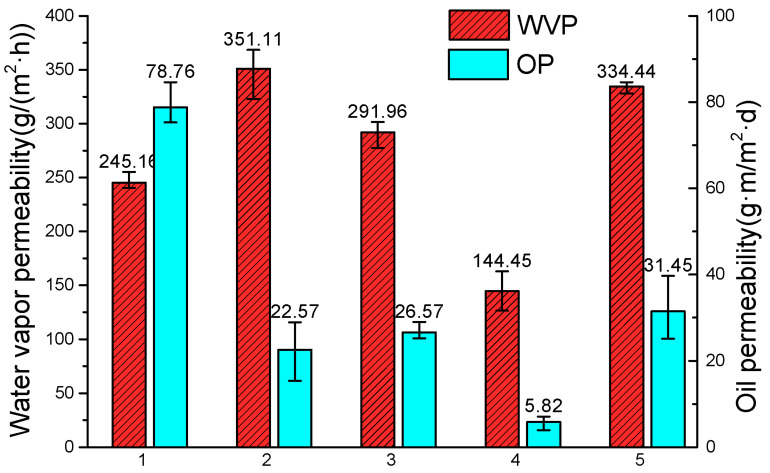
Gas permeability of SPI and fractioned product films: (1) SPI, (2) lower protein, (3) upper protein, (4) 11S globulin, and (5) 7S globulin.

**Figure 8 materials-14-05436-f008:**
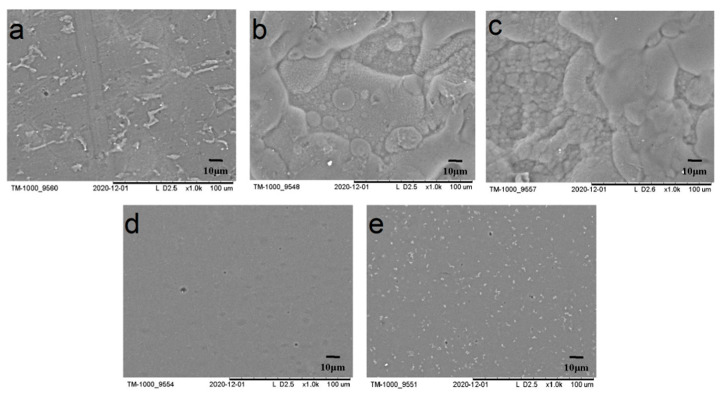
SEM images of SPI and fractioned films: (**a**) SPI, (**b**) upper protein, (**c**) lower protein, (**d**) 11S globulin, and (**e**) 7S globulin.

**Table 1 materials-14-05436-t001:** Yields and contact angles of the samples.

	Weight (g/24)	Yield (%)	Contact Angle (°)
SPI	-	-	47.7 ± 0.5
Upper protein	1.50 ± 0.26	6.25 ± 1.10	60.9 ± 0.6
Lower protein	13.96 ± 1.04	58.19 ± 4.32	57.0 ± 0.5
11S	3.22 ± 0.66	13.42 ± 2.75	36.2 ± 0.8
7S	4.45 ± 0.42	18.57 ± 1.75	25.7 ± 0.3

## Data Availability

All data are freely available.

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
