# Peer review of "Preparation and Properties of Fractionated Soybean Protein Isolate Films"

_materials, 2021, doi:10.3390/ma14185436_

Round 1
Reviewer 1 Report
The authors proposed the preparation of soybean protein isolate products, thin film fabrication, and the evaluation of thin film properties. This study was highly attractive in the point of natural film fabrication. However, in this present version, the manuscript is necessary to improve in various points prior to reconsideration.
- The submitted manuscript must be followed the template of Materials. Please recheck about the instruction to authors in this link: https://www.mdpi.com/journal/materials/instructions
- Abstract was very confused and need to revise all. It would be better to mention about the objective, the novelty of this study, and the impact results from this study.
- Introduction part was too short and also need to revise all. It would be better to mention in these following points;
- Background about edible packaging films: What is it? Why it became to be a research hotspot?
- The existing research about the edible packaging film: Raw materials, process and technology of thin film fabrication, advantage and disadvantage of these existing edible packaging films. The challenging points which need to be solved.
- The motivation of this study: Why do you choose “soybean protein isolate” as the raw material? What are the attractive points over other kinds of raw material? And the comparison between this centrifugal precipitation and thin film fabrication compared with the existing method of edible packaging film fabrication.
- The last paragraph of the introduction should be mentioned about the detailed of this study included about the objective and the expectation of the obtained results.
- From the introduction part, it was mentioned that the protein content of SPI composed of 2S, 7S, 11S, and 15S globulin. Why do you selected to consider only 7S and 11S in this study?
- About the edible packaging films, are there possible to degrade during use or contaminate with the food? How about in this study?
- Materials and methods;
- Materials: All materials and chemicals used in this study should be listed. Also, the chemical formula, full name of company and its location should be mentioned.
- The listed of instruments should be mentioned in section 1.2.4. Also, the full name of company and location should be added.
- The abbreviation should be firstly mentioned in the full name prior to use as the short name such as PBS, SDS-PAGE.
- The detailed condition of each characterization method should be mentioned all.
- The “film solution” for the rheological testing should be further explained about the sample preparation prior to testing.
- The characterization condition of contact angle measurement should be mentioned such as water volume and testing temperature.
- The equation of tensile strength and elongation at break should be mentioned as Equation (1) and (2) as the template of journal.
- The equation of fracture elongation should be rechecked. It would be better to mention about the standard distance of sample (L0 = ?? mm). Also, L which is the distance of elongated specimen should not be the “standard distance”.
- The calculation of water vapor permeability (WVP) and oil permeability (OP) should be added.
- The characterization condition of SEM should be mentioned such as voltage, testing temperature, magnification, mode of characterization, and the condition of gold sputtering such as amount and time.
- Discussion and Results -> Results and discussion
- Figure 1: the interpretation of subunits and the marker in the left-hand side should be included.
- Table 1: The calculation method of weight (g/24 g) and yield (%) should be mentioned in the materials and method section.
- From the sentence of “so the selection of separation methods is very important.”, What is the relationship with the centrifugal precipitation method in this study? Is it the best method? How can it compare with other existing method? How about the yield of SPI fractionated samples of this method compared with other methods?
- What is the relationship between the rheological results and other properties of each sample?
- Figure 2; sheer rate (1/s) -> Shear rate (s-1)
- Figure 3; it would be better to write down the water contact angle of each sample at the top of each droplet images.
- According to the targeted application of the obtained thin film products, what is the main expectation between hydrophilicity and hydrophobicity? What is the reason why the contact angles are different between each sample?
- Figure 4; elongation at break of 70.47% belongs to the upper protein film, not the lower protein film in the sentences of section 2.4.
- What is the most important for the targeted application of the obtained film between tensile strength and elongation at break?
- The difference of tensile properties of lower and upper protein should be also discussed. Why tensile strength was similar but the elongation at break was totally different?
- Figure 5; FTIR spectrum -> FTIR spectra
- Figure 5; The number of wavelengths in the graph was too small and difficult to read. It would be better to draw a straight line of the mentioned wavenumber in order to easier comparison between each sample.
- How can FTIR results show the differentiate of molecular weight of proteins? It would be better to add and discuss the results of molecular weight and molecular weight distribution.
- Why 11S globulin has low water vapor and oil permeability as compared to other samples even it has higher hydrophilicity (WCA = 36.2 deg) than 7S globulin (WCA = 25.7 deg)?
- Figure 7; it was mentioned that 7S and 11S globulin showed the smooth surface which was beneficial to film extension. However, the elongation at break of 7S globulin was lower than the upper and lower protein (Figure 4). Also, the morphology of SPI looks smoother than the upper and lower protein but its elongation at break was the lowest. The reason of this confliction between surface morphology and the elongation at break should be discussed.
- Figure 7; the images and the scale bar should be enlarged for better understanding.
- The ordering of the samples should be the same in all Figures and Tables for easy comparison between each results. For example, a) SPI, b) lower protein, c) upper protein, d) 7S globulin, and e) 11S globulin.
- Conclusion should be revised. It would be better to emphasize the success of this study. What is the novelty of this study? What is the best samples and the best properties? What is the targeted application and how the best obtained product can be used and solve the exist challenging points? What are the future prospects of this study?
- References part should be revised and followed the style of Materials.
- English writing should be improved and rechecked by English native speaker throughout manuscript.
Author Response
Dear reviewer,
The attachment is the respose of the comments.
Best regards!
Yu Zuolong

Reviewer 2 Report
The article is written correctly. May be of interest to the readers of the magazine. I suggest introducing minor additions: 1. SEM images of the received films are missing. They are only from optical microscopy. I think SEM would be good to complement this. 2. There are no DSC measurements of the films obtained, and in the case of polyols this is more important than presenting FTIR measurements. 3. The authors did not propose a specific application for the received films. After these amendments are made, the article may be published.
Author Response

(The authors gave the same response as above.)

Reviewer 3 Report
Dear authors,
The paper ‘’Preparation and Properties of Fractionated Soybean Protein Isolate Films’’ provided first hand findings about the functionalities of SPI sub-fractions to be utilized in packaging films, however the following points should be clearly addressed for further processing of your paper:
- It should be specified in the Abstract that which fraction or fractions could show better results.
- The introduction part needs more literature review, hence the following could help you to write this part in a better state:
- Chen, H., et al. (2019). Application of protein-based films and coatings for food packaging: A review. Polymers, 11(12), 2039.
- Garavand, F., et al. (2020). A comprehensive review on the nanocomposites loaded with chitosan nanoparticles for food packaging. Critical reviews in food science and nutrition, 1-34.
- Mihalca, Vlad, et al. "Protein-based films and coatings for food industry applications." Polymers 13, no. 5 (2021): 769.
- Bahrami, R., et al. (2020). Modification and improvement of biodegradable packaging films by cold plasma; a critical review. Critical Reviews in Food Science and Nutrition, 1-15.
- The full details of the materials suppliers should be added to the materials section.
- The full details of manufacturers of equipment should be mentioned in the text as well.
- The 1.1. title should be Materials and equipment not only Materials
- Who much mM sodium sulfite and 5 mM anhydrous magnesium chloride to the extract to fractionate 11S?
- For all conducted preparations and processing the full details on time, temperature, speed, etc. should be provided, e.g. 1.2.2-please apply it to whole text.
- Change the rpm units to g in the whole text.
- The applied denaturing incubation temperature looks higher than the standards, it is normally 70C for 10 min, or 90C for 5 min, please justify the reason?
- Rewrite the 1.2.3. Section, it seems confusing.
- The drying condition for films also sounds high, for how long you used this temperature?
- Why you set the rheometer constant temperature on 80C? Room temperature makes more sense.
- You can separate the characterisation techniques and provide further details on each.
- What do you mean by… the lower protein was not dissolved???, but similar bands could be observed with the other samples, the sample must be dissolved in the provided reagents, the reason should be due to its lower pro content.
- The images of the prepared films should be presented separately with a better quality.
- What is the reason behind the viscosity drop in 11S samples in higher shear rates?
- The following paper could help you with a better interpretation of contact angle:
- Mirzaei-Mohkam, et al. (2020). Physical, mechanical, thermal and structural characteristics of nanoencapsulated vitamin E loaded carboxymethyl cellulose films. Progress in Organic Coatings, 138, 105383.
- Are any of the generated peaks could be associated with the interaction of proteins and other film ingredients? If so, please discuss and justify
- The references are not arranged according to the journal guidelines, please read the instructions carefully.
Author Response

(The authors gave the same response as above.)

Round 2
Reviewer 1 Report
The manuscript was significantly improved. Thank you very much for the revision.
Reviewer 2 Report
Acept
Reviewer 3 Report
Almost all mentioned comments are addressed.